

# The role of CEMIP in cancers and its transcriptional and post-transcriptional regulation

Song Guo, Yunfei Guo, Yuanyuan Chen, Shuaishuai Cui, Chunmei Zhang and Dahu Chen

Shandong University of Technology, School of Life Sciences and Medicine, Zibo, Shandong, China

## ABSTRACT

CEMIP is a protein known for inducing cell migration and binding to hyaluronic acid. Functioning as a hyaluronidase, CEMIP primarily facilitates the breakdown of the extracellular matrix component, hyaluronic acid, thereby regulating various signaling pathways. Recent evidence has highlighted the significant role of CEMIP in different cancers, associating it with diverse pathological states. While identified as a biomarker for several diseases, CEMIP's mechanism in cancer seems distinct. Accumulating data suggests that CEMIP expression is triggered by chemical modifications to itself and other influencing factors. Transcriptionally, chemical alterations to the CEMIP promoter and involvement of transcription factors such as AP-1, HIF, and NF-κB regulate CEMIP levels. Similarly, specific miRNAs have been found to post-transcriptionally regulate CEMIP. This review provides a comprehensive summary of CEMIP's role in various cancers and explores how both transcriptional and post-transcriptional mechanisms control its expression.

## INTRODUCTION

Cell migration-inducing and hyaluronan-binding protein (CEMIP), also known as KIAA1199 or HYBID, is a multifunctional protein involved in cell migration promotion and hyaluronic acid (HA) binding. It was initially identified in 1999 from a cDNA library screen (*Kikuno et al., 2002*). CEMIP is recognized by several aliases, including colon cancer secretory protein 1 (CCP1), transmembrane protein 2-like (TMEM2L), and hyaluronan-binding protein involved in hyaluronan depolymerization (HYBID). The CEMIP gene is located on human chromosome 15q25.1 and encodes 1361 amino acids. Operating as a pro-vascular and pro-inflammatory factor, CEMIP is widely expressed in various normal human tissues such as the brain, lung, and pancreas (*Michishita et al., 2006*). The subcellular localization and functions of CEMIP vary depending on the cell type. In colon cancer cells, CEMIP is predominantly found in the cytoplasm (*Birkenkamp-Demtroder et al., 2011*), nucleus (*Birkenkamp-Demtroder et al., 2011*), and cell membrane (*Sabates-Bellver et al., 2007*). Its presence in these locations facilitates proliferation, apoptosis, metastasis, and invasion *in vitro* and *in vivo*, suggesting its involvement in tumorigenesis and metastasis. In breast cancer, cytoplasmic CEMIP promotes cell proliferation, migration, and invasion

Corresponding author
Dahu Chen, dahuchen@sdut.edu.cn

**Figure 1  Schematic diagram of secondary structure of CEMIP.** CEMIP protein is composed of several domains: SP, G8, GG, WxxW repeats and PbH1 repeats. Among them, the GB domain has eight glycines, and is expected to contain one α-helix and 10 β-strands. GB domain has been reported to bind with EGFR, PlexinA2, ANXA1 and other factors to trigger different signals. Two GG structural domains, each consisting of 7 β-strands and 2 α-helices. The second GG domain can interact with coatomer protein complex α-subunit (COPA) and glycogen phosphorylase kinase β-subunit (PHKB). The function of WxxW repeats is unknown, but its WxxW sequence is highly conserved. PbH1 repeats consists of four parallel β -repeats sequences. It is known that PbH1 repeats domain can bind to O-GLCNAC transfer (OGT), but its specific function is unknown.

(*Jami et al., 2014*). Structurally, CEMIP consists of one G8 domain, two GG domains, four PBH1 domains, and several N-linked glycosylation sites (*Guo et al., 2006*; *He et al., 2006*; *Yoshida et al., 2014*). Proteins containing G8 domains are typically predicted to localize in the nucleus and cytoplasm, with the G8 domain of CEMIP possibly influencing its subcellular localization (*Liu et al., 2021b*). The precise role of GG domains in hyaluronic acid binding and degradation remains unclear (*Yoshida et al., 2013*). Refer to Fig. 1 for the secondary structure of CEMIP.

In addition to its involvement in cancer, CEMIP interacts with various signaling molecules such as transcription factors, EGFR, and the Wnt/β-catenin pathway, participating in essential cellular activities. CEMIP expression can activate epithelial-mesenchymal transition (EMT)-related signaling pathways (*Wang et al., 2019a*). Studies indicate that classical and non-canonical Wnt signaling promote tissue-specific tumor cell metastasis, tightly linking Wnt signaling to colorectal cancer (CRC) development (*Zhan, Rindtorff & Boutros, 2017*). Furthermore, CEMIP expression independently predicts prognosis in CRC (*Xu et al., 2015*). In specific contexts, CEMIP activates Wnt/β-catenin signaling (*Chen et al., 2021a*; *Sabates-Bellver et al., 2007*; *Xue et al., 2020*) and may stimulate PI3K/AKT pathway molecules, regulating ovarian cancer (*Shen et al., 2019*). Notably, acting as a hyaluronidase, CEMIP breaks down high molecular weight hyaluronic acid in the extracellular matrix into low molecular weight HA and oligosaccharides (*Chanmee, Ontong & Itano, 2016*). This process involves the release of HA-coated vesicles into the extracellular space and significantly contributes to normal HA catabolism in arthritis (*Deroyer et al., 2019*). CEMIP's hyaluronidase activity might promote tumorigenesis through multiple mechanisms, as the low molecular weight HA produced by CEMIP can enhance metastasis and invasion (*Dang et al., 2013*; *Gao et al., 2010*; *Schmaus, Bauer & Sleeman, 2014*; *Sugahara et al., 2003*).

These studies collectively highlight the diverse functions of CEMIP, emphasizing the need for further exploration. Substantial evidence supports CEMIP's potential as a biomarker for various diseases, with most preclinical studies showing an association between abnormal CEMIP expression and disease onset. Moreover, CEMIP is implicated in the pathogenesis of several human cancers, including prostate, breast, and colorectal cancer.

While investigating the tumor biological functions of CEMIP, attention has shifted toward understanding the regulation of CEMIP expression itself. Currently, research on CEMIP's transcriptional and post-transcriptional regulation remains limited. Various factors and chemical modifications of the CEMIP gene contribute to both transcriptional and post-transcriptional regulation. Existing studies indicate that CEMIP transcription can be induced by different transcription factors and microRNAs.

This review primarily focuses on delineating CEMIP's role in human cancer and its association with the regulation of CEMIP expression. Additionally, it delves into the analysis of how transcriptional and post-transcriptional control mechanisms of CEMIP influence its expression, consequently leading to specific pathological outcomes. The intended audience for this article comprises cancer researchers and peers within related biomedical disciplines, particularly those dedicated to CEMIP research and the development of new cancer drugs targeting CEMIP.

## SURVEY METHODOLOGY

This article presents findings derived from a comprehensive longitudinal query analysis of the MEDLINE database hosted on PubMed and Web of Science. The data collection process involved referencing all pertinent literature using specific keywords or their combinations: "CEMIP," "CEMIP and transcription factors," "CEMIP and chemical modification," "CEMIP and transcriptional regulation," "CEMIP and miRNAs," "CEMIP and post-transcriptional regulation," and "CEMIP and cancers." The literature citations aim for a comprehensive and impartial coverage of the topic.

## THE ROLE OF CEMIP IN CANCERS

In this section, we aim to consolidate pathophysiological evidence supporting CEMIP's involvement in various cancer processes.

### CEMIP and prostate cancer

Prostate cancer stands as a prevalent malignant tumor in adult men. Investigations reveal a significant increase in CEMIP expression within human prostate cancer cells compared to control cells. Mechanistically, CEMIP's involvement in angiogenesis is evident through increased semaphoring 3A (sema3A) and decreased levels of vascular endothelial growth factor A (VEGFA), vascular endothelial cadherin (VE-cadherin), phosphorylated erythropoietin-producing hepatocellular A2 (EphA2), and hyaluronic acid polymers (*Luo et al., 2022*).

Anoikis, a form of programmed cell death activated upon cell detachment from the extracellular matrix (ECM), undergoes alteration in cancer cell metabolism, potentially enhancing survival and metastasis. In prostate cancer cells, CEMIP overexpression, triggered by AMPK/GSK3 β/β-catenin signaling, may promote migration, invasion, and resistance to anoikis *via* PDK4-mediated metabolic reprogramming (*Zhang et al., 2018*). Detached cells may also experience non-apoptotic cell death due to rectifying metabolic deficiencies (*Hawk et al., 2018*). Escaping anoikis or other non-apoptotic cell death is
essential for cancer cells to gain metastatic capabilities. Additionally, CEMIP involvement in ferroptosis—a form of programmed cell death induced by iron—has been observed in prostate cancer cells. Upregulated CEMIP facilitates ferroptosis resistance during ECM detachment by promoting cystine uptake. Silencing CEMIP inhibits ferroptosis and cystine uptake (*Liu et al., 2022*).

## CEMIP and colon cancer

Studies indicate a significant increase in CEMIP expression in colon cancer cells compared to controls, correlating closely with poor prognosis in colorectal cancer (CRC) (*Tiwari et al., 2013*; *Weng et al., 2023*; *Wu et al., 2021*). *Tiwari et al. (2013)* demonstrates CEMIP as one of the Wnt signal targets, with its upregulation inhibiting cell growth by negatively regulating Wnt signal transduction. However, recent findings by Birkenkamp-Demtroder et al. suggest that shRNA-mediated repression of CEMIP expression attenuates Wnt signaling in parental SW480 cells (*Tiwari et al., 2013*). Further research is needed to clarify CEMIP's exact role in regulating Wnt signaling.

Apart from its direct regulation of Wnt signaling, CEMIP induction through a beta-catenin and FRA-1 dependent pathway in BRAF v600E mutated colorectal cancers has been observed (*Duong et al., 2018*). CEMIP's interaction with MEK-1 sustains ERK1/2 activation, maintaining c-Myc protein levels and providing metabolic advantages (sustaining amino acid synthesis) in these cells. Silencing CEMIP decreases ERK1/2 signaling and c-Myc protein levels, signifying CEMIP's role in the cross-talk between WNT and MAPK signaling in colorectal cancers.

Furthermore, CEMIP's impact on major histocompatibility complex class I (MHC-I) levels in colorectal cancer cells is noteworthy. CEMIP decreases MHC-I expression at the protein level by promoting MHC-I internalization *via* clathrin-mediated endocytosis, leading to MHC-I degradation in lysosomes. This downregulation inhibits the cytotoxicity of CD8+ T cells, decreasing their antitumor activities (*Zhang et al., 2023*). CEMIP's downregulation of MHC-I levels on the cell surface plays a crucial role in immune escape, suggesting CEMIP inhibition as a potential approach in cancer immunotherapy.

CEMIP also influences colorectal cancer cell behavior. Knocking out CEMIP inhibits CRC cell proliferation and induces cell G1 arrest (*Liang et al., 2018*). Additionally, CEMIP activates CDC42/MAPK pathway-mediated EMT by enhancing GRAF1 degradation. GRAF1 plays an essential role in CEMIP-mediated CRC migration and invasion (*Xu et al., 2023*).

Moreover, CEMIP functions as an O-GlcNAc transferase (OGT) adapter protein. Its interaction with OGT and β-catenin increases O-GlcNAcylation of β-catenin, enhancing its translocation into the nucleus. Nuclear β-catenin enhances CEMIP transcription, leading to overexpression of glutaminase 1 and glutamine transporters (SLC1A5 and SLC38A2). SLC1A5 and SLC38A2 facilitate glutamine transport into cells, which has been linked to cancer metastasis. Targeting glutamine metabolism combined with CEMIP modulation significantly reduces CRC metastasis (*Hua et al., 2021*).

These findings underscore CEMIP's value as a prognostic biomarker and a potential therapeutic target for colorectal cancer.

## CEMIP and pancreatic cancer

Pancreatic cancer represents a highly malignant tumor, primarily originating from adenocarcinoma in ductal cells. Pancreatic ductal adenocarcinoma (PDAC) is characterized by a dense desmoplastic stroma enriched with hyaluronan (HA). Accumulation of HA in PDAC plays a crucial role in cancer invasion and metastasis. Acting as a hyaluronidase, CEMIP degrades hyaluronic acid in the extracellular matrix, reducing its viscosity and promoting cell movement. Comparative analysis reveals a significant increase in CEMIP mRNA levels in pancreatic ductal adenocarcinoma (PDAC) cells compared to non-tumor cells, correlating with poor survival among PDAC patients (*Koga et al., 2017*) .

Hypoxia amplifies the migratory ability of PDAC cells by upregulating CEMIP expression. CEMIP catalyzes HA degradation, generating low-molecular-weight HA, fostering a favorable microenvironment for PDAC progression (*Oba et al., 2021*). *In vitro* studies indicate that CEMIP contributes to the proliferation and migration of PDAC cells (*Kohi et al., 2017*). The pro-inflammatory cytokine interleukin-1ßincreases CEMIP transcription, enhancing PDAC cell migration. Given the challenges in early-stage pancreatic cancer diagnosis, carbohydrate antigen (CA) 19-9, a pancreatic cancer biomarker, has limited sensitivity and specificity for general screening (*van den Bosch et al., 1996*).

CEMIP's overexpression in pancreatic cancer correlates with poor survival and advanced stages. Combining CA19-9 data with CEMIP significantly enhances the accuracy of pancreatic cancer detection by complementing each other, thereby improving diagnosis sensitivity and specificity. This combined approach facilitates early-stage pancreatic cancer detection, crucial for enhancing patient survival and treatment outcomes (*Lee et al., 2018*). Hence, CA19-9 and CEMIP stand as promising biomarkers whose combined use could enhance early pancreatic cancer detection. However, further research is essential to validate their performance in larger, prospective studies and ascertain their role in screening high-risk populations.

## CEMIP and lung cancer

Histopathologically, lung cancer divides into non-small cell lung cancer (NSCLC) and small cell lung cancer (SCLC). Current research demonstrates significantly elevated CEMIP mRNA and protein levels in NSCLC or SCLC cells compared to adjacent normal tissues. High CEMIP expression correlates with poor survival among NSCLC patients (*Tang et al., 2019*). Manipulating CEMIP levels in NSCLC lines—either knocking it out or overexpressing it—led to respective decreases or increases in the expression of epithelial-mesenchymal transition (EMT) marker genes, mediated by PI3K-Akt signaling (*Tang et al., 2019*). CEMIP promotes SCLC cell migration, proliferation and invasion (*Li et al., 2020*; *Li et al., 2023*; *Mo et al., 2023*).

Moreover, CEMIP's hyaluronidase activity leads to the depolymerization of high molecular weight hyaluronic acid into low molecular weight forms. The accumulation of low molecular weight hyaluronic acid activates its receptor TLR2, recruiting c-Src and activating ERK1/2 signaling, thus promoting F-actin rearrangement and SCLC cell migration and invasion (*Li et al., 2023*).

Studies also reveal that CEMIP disrupts the interaction between FBXW7, an E3 ubiquitin ligase, and c-Myc in SCLC cells (*Mo et al., 2023*). This interference reduces the ubiquitination level of c-Myc, leading to its stabilization and increased nuclear accumulation. Through its indirect regulation of c-Myc, CEMIP promotes glutamine-dependent proliferation in SCLC cells.

## CEMIP and breast cancer

Breast cancer, a frequently diagnosed malignancy in women globally, often presents a challenging prognosis. Current research highlights a significant increase in CEMIP expression in breast cancer tissues compared to normal tissues. Elevated CEMIP levels correlate with decreased overall survival among breast cancer patients (*Dong et al., 2021*; *Jami et al., 2014*; *Kuscu et al., 2012*; *Xue et al., 2022*). CEMIP promotes the proliferation and migration of breast cancer cells by activating STAT3 signaling (*Chen, Li & Zhang, 2021b*).

Within the tumor microenvironment (TME), CEMIP potentially plays a carcinogenic role by contributing to extracellular matrix formation, elevating cancer-associated fibroblast (CAF), M2 macrophage, and neutrophil infiltration, while reducing CD8+ T cell infiltration (*Dong et al., 2021*). Recent studies have also linked CEMIP expression in breast cancer to various immune-related molecules, encompassing both immune inhibitors and stimulators, alongside MHC molecules.

In addition to its direct tumor-promoting role, CEMIP might regulate other tumor proteins to enhance its carcinogenic potential. For instance, binding immunoglobulin protein (BiP), known to drive cancer progression and metastasis, undergoes upregulation in breast cancer. CEMIP activation of the BiP promoter upregulates BiP transcript and protein levels in breast cancer cell lines (*Banach et al., 2019*).

## CEMIP and gastric cancer

Gastric cancer, the second most prevalent malignant tumor with high mortality after lung cancer (*Jemal et al., 2011*), presents multifactorial causation, with Helicobacter pylori infection being a primary factor. Gastric cancer subtypes are classified based on histology, aiding in subtype identification, prognosis, and treatment response (*Lauren, 1965*). While CA19-9 and CA72-4 serve as biomarkers for gastric cancer, emerging studies also indicate CEMIP's relevance as a diagnostic marker.

Upregulated CEMIP expression in gastric cancer tissues correlates with poor prognosis (*Jia et al., 2017*; *Matsuzaki et al., 2009*). Studies reveal CEMIP's involvement in Wnt signaling and the EMT process in gastric cancer. Its reduced expression weakens β-catenin's ability to transmit Wnt signals, influencing downstream target genes like c-myc, cyclin D1, and EMT markers (*Jia et al., 2017*). Additionally, CEMIP mediates gastric cancer progression, involving HIF-1 α (*Mi et al., 2023*).

Targeting CEMIP emerges as a promising therapeutic strategy for gastric cancer, considering its role in promoting tumor progression, metastasis, and its involvement in multiple signaling pathways. Natural compounds like curcumin, resveratrol, and quercetin have demonstrated the suppression of CEMIP expression and activity in gastric cancer

cells, inhibiting glycolysis, migration, and invasion (*Zhao et al., 2022*) . Furthermore, angiogenesis-targeted therapies like ramucirumab and apatinib may indirectly modulate CEMIP levels by influencing the tumor microenvironment (*Nakayama & Takahari, 2023*).

However, further research is crucial to elucidate the specific mechanisms and clinical efficacy of targeting CEMIP in gastric cancer.

## CEMIP in various other cancers

CEMIP protein levels significantly escalate in laryngeal squamous cell carcinoma, correlating with adverse clinicopathological parameters (*Huang et al., 2020*). In hepatocellular carcinoma (HCC), high CEMIP expression promotes tumor growth and correlates with reduced patient survival (*Jiang et al., 2018*; *Xu et al., 2019*). In glioblastoma, CEMIP downregulation decreases cell proliferation and migration, while its expression aids macrophage migration (*Tsuji et al., 2021*). Moreover, CEMIP's overexpression in osteosarcoma cells stimulates growth and metastasis by activating Notch signaling (*Cheng et al., 2022*). However, in chondrosarcoma, CEMIP expression exhibits anti-tumor effects on tumor growth *in vivo* (*Koike et al., 2020*). In cholangiocarcinoma cells, high CEMIP expression enhances growth and metastasis, highlighting its oncogenic role (*Zhai et al., 2020*)

In summary, current evidence emphasizes the pivotal role of CEMIP in human cancer development and progression, warranting further comprehensive exploration of its diverse functions in cancer-related processes.

# CEMIP: TRANSCRIPTIONAL AND POST-TRANSCRIPTIONAL REGULATION

## Transcriptional regulation

Transcriptional regulation governs gene expression by overseeing the process of DNA transcription into RNA. This orchestration allows cells to respond adequately to developmental and environmental cues by activating or repressing genes as required. This regulation comprises multiple facets, including control by transcription factors and epigenetic modifications like DNA methylation and histone modifications. Present investigations into CEMIP transcriptional regulation have primarily centered around the control by transcription factors. This overview will encapsulate advancements in understanding how transcription factors regulate CEMIP. Additionally, a brief insight into the epigenetic regulation of CEMIP's transcription will be provided.

### AP-1

The transcription factor Activator Protein-1 (AP-1) is a dimeric protein complex consisting of Jun, Fos, or ATF subunits (*Karin, Liu & Zandi, 1997*). Known for regulating numerous target genes, AP-1's activity is modulated by various signaling pathways. For instance, the absence of bone morphogenetic protein 6 (BMP6) suppresses AP-1 transcriptional activity (*Lu et al., 2023*). In different cancers, AP-1's effects vary depending on context (*Mathas et al., 2002*; *Zhao et al., 2014*). Engaging with signaling pathways like JNK, ERK, and Notch, AP-1 participates in diverse biological processes. External stimuli such as

growth factors or UV radiation activate AP-1, enabling it to bind to specific target gene promoter or enhancer regions, thereby boosting their transcription (*Lee, Mitchell & Tjian, 1987*). In breast cancer, AP-1 binds to the CEMIP promoter, activating its transcription, and consequently promoting cancer cell growth (*Kuscu et al., 2012*). Conversely, hindering AP-1's transcriptional activity might lower CEMIP expression and exacerbate myocardial fibrosis (*Lu et al., 2023*). Additionally, the CEMIP promoter harbors a potential binding site for JUN, an AP-1 subunit, indicating its role as a potential CEMIP transcriptional activator (*Dong et al., 2021*). These studies underscore the significance of AP-1 as a transcriptional regulator of CEMIP.

### NF-κB

NF-κB, a family of transcription factors, holds pivotal roles in numerous biological processes and tumorigenesis. Comprising five members—NF-κB1 (p50), NF-κB2 (p52), RelA (p65), RelB, and c-Rel—this family commonly forms dimers such as p65/p50 and RelB/p52. NF-κB proteins contain a Rel homology domain (RHD) that facilitates their dimerization. Dimers like p65/p50 and RelA/c-Rel bind to promoter sequences, thereby regulating the expression of target genes. Additionally, the co-activator Bcl-3 modulates NF-κB-mediated transcription. NF-κB activation is meticulously regulated, with NF-κB1, RelA, and c-Rel being activated *via* classical pathways, while NF-κB2 and RelB use alternative pathways. Notably, the carcinogenic potential of NF-κB predominantly hinges on the activity of p50 and p65 (*McKeithan et al., 1987*). Functionally, NF-κB governs cell proliferation and invasion, demonstrated by its inhibition of proliferation in keratinocytes (*Hinata et al., 2003*).

Activated NF-κB robustly enhances gene transcription. Within the CEMIP promoter, multiple distant NF-κB sites exist, although their specific function remains unclear. Nevertheless, NF-κB elements have the capacity to bolster CEMIP promoter activity (*Kuscu et al., 2012*). Moreover, the endogenous B cell leukemia-3 (Bcl-3) protein is recruited to the CEMIP NF-κB site and is essential for histone acetylation (*Shostak et al., 2014*). Consequently, NF-κB transcription factors play a pivotal role in activating CEMIP transcription and modulating its biological functions.

### HIF

Mammalian cell metabolism and energy production rely on oxygen, a key driver that also fosters solid tumor growth. Hypoxic cells adapt through mechanisms regulated by the transcription factor hypoxia-inducible factor (HIF), predominantly *via* transcriptional and post-transcriptional means. HIF operates as a heterodimer, comprised of either HIF-1 α or HIF-2 α and HIF-1 β/ARNT subunits (*Wang et al., 1995*). While three HIF-α subunits exist—HIF-1 α, HIF-2 α, and HIF-3 α—HIF-1 α and HIF-2 α serve as the primary activators of hypoxia-induced genes (*Ema et al., 1997*; *Semenza & Wang, 1992*). HIF-1 α exhibits widespread expression, whereas HIF-2 α expression is more tissue-specific (*Ema et al., 1997*). When confronted with hypoxic conditions, HIF-1 α and HIF-1 β form an active HIF complex that binds to hypoxia response elements (HREs) within target genes (*Wenger, Stiehl & Camenisch, 2005*), thereby inducing the upregulation of numerous hypoxia-associated genes (*Rankin & Giaccia, 2016*).
In colon cancer, CEMIP mRNA and protein levels surge under hypoxic conditions compared to normoxic conditions. Bioinformatics analyses have unveiled potential HREs in the CEMIP sense strand from positions 125 to 120 bp. Although the depletion of HREs did not entirely suppress CEMIP promoter activity, presumably due to other cis-elements, HIF can directly prompt CEMIP transcription, partially mediated through HIF-2 α (*Evensen et al., 2015*). Conversely, a separate study did not observe direct HIF-1 α induction of CEMIP; however, it noted a strong correlation between CEMIP expression and HIF-1 α levels in hypoxic hepatocellular carcinoma (*Wenger, Stiehl & Camenisch, 2005*). These findings collectively underscore the significant role of HIF in governing CEMIP transcriptional regulation under hypoxic conditions

### Sp1

Sp1, the pioneering member of the Sp family of transcription factors encompassing Sp2, Sp3, and Sp4, is recognized for its C2H2-type zinc finger structures (*Black, Black & Azizkhan-Clifford, 2001*; *Bouwman & Philipsen, 2002*; *Kaczynski, Cook & Urrutia, 2003*). Sp1 and Sp3 share highly homologous DNA binding domains, with Sp1 displaying three zinc fingers, each imparting distinct binding preferences, while the first finger confers high sequence specificity (*Kriwacki et al., 1992*; *Thiesen & Schröder, 1991*). Initially identified as a promoter-specific binding factor essential for the transcription of the SV40 major immediate early (*Dynan & Tjian, 1983a*; *Dynan & Tjian, 1983b*). Sp1 and the long isoform of Sp3 contain two transactivation subdomains, A and B, where the D domain of the Sp1 C-terminus is crucial for synergistic activation. Known for its regulatory roles in metabolism, proliferation, apoptosis, and development, Sp1 expression is often heightened in cancer cells, correlating with poorer prognoses in cancers like colorectal, gastric, and lung (*Chuang et al., 2009*; *Davie et al., 2008*; *Kong et al., 2010*).

These Sp factors typically prefer binding to GC box motifs. By binding to GC-rich sites within target gene promoters, Sp1 can activate their transcriptional activity (*Hagen et al., 1995*; *Kingsley & Winoto, 1992*; *Thiesen & Bach, 1990*). Despite the presence of a GC box at positions 248/243 within the CEMIP promoter, a study revealed that removing this specific GC box did not affect CEMIP transcription, creating an apparent contradiction (*Kuscu et al., 2012*). Therefore, the exact role of Sp1 in inducing CEMIP transcription warrants further investigation.

### β-catenin

β-catenin, a highly conserved and multifunctional protein, possesses a central region featuring 12 imperfect Armadillo repeats flanked by distinctive N-terminal and C-terminal domains (*Valenta, Hausmann & Basler, 2012*). Functionally, *β*-catenin serves as both a structural component within cadherin-based cell junctions and a pivotal nuclear effector of canonical Wnt signaling, controlling cell proliferation. As part of the Wnt pathway, nuclear β-catenin activates the transcription of target genes like c-myc and cyclin D1 (*Reiss et al., 2005*).

In colorectal cancer cells, the β-catenin/TCF4 complex orchestrates CEMIP transcription upstream of the CEMIP promoter (*Duong et al., 2018*; *Sabates-Bellver et al., 2007*). Consequently, inhibiting the β-catenin/TCF complex significantly reduces CEMIP

levels (*Sabates-Bellver et al., 2007*). Nuclear β-catenin accumulation augments CEMIP transcription. Upon interaction with OGT and β-catenin, CEMIP heightens O-glycosylation of β-catenin, facilitating its nuclear translocation from the cell membrane (*Hua et al., 2021*). These observations hint at a positive regulatory cycle between CEMIP and β-catenin, potentially affecting Wnt/*β*-catenin signaling (*Duong et al., 2018*).

### ATF4

Activated Transcription Factor 4 (ATF4) belongs to the basic leucine zipper (bZIP) family of transcription factors. These bZIP proteins form homo- or heterodimers through their leucine zipper domain, regulating transcription once dimerized (*Landschulz, Johnson & McKnight, 1988*; *Landschulz, Johnson & McKnight, 1989*; *Vinson, Sigler & McKnight, 1989*). ATF4 serves as a crucial subunit within many heterodimeric bZIP complexes but lacks individual transcriptional activity (*Ebert et al., 2022*). The transcriptional functions of ATF4 are contingent upon its heterodimerization with other bZIP proteins.

CEMIP has been identified as a target gene of ATF4-containing heterodimers. Through chromatin immunoprecipitation analysis, several potential ATF4 binding sites have been revealed within the CEMIP promoter. Subsequent experiments have conclusively demonstrated that ATF4 induces CEMIP transcription by augmenting CEMIP promoter activity and expression (*Yu et al., 2022*)

## EPIGENETIC MODIFICATION-METHYLATION

DNA methylation stands as a crucial epigenetic mechanism, where DNA methyltransferases transfer methyl groups from S-adenosylmethionine to the C5 position of cytosine, forming 5-methylcytosine (*Robertson & Jones, 2000*). This mechanism regulates gene expression at the transcriptional level. Typically, CpG islands within gene promoters and exons remain unmethylated under normal conditions. However, in cancer cells, genome-wide hypomethylation is evident, while specific CpG islands undergo hypermethylation. While DNA methylation is generally perceived to inhibit transcription, recent evidence suggests that DNA demethylation can upregulate cancer-related genes.

In a study involving breast cancer cell lines MCF-7 and MDA-MB-231 treated with demethylation agents, distinct responses in CEMIP expression were observed. Notably, CEMIP expression significantly increased in MCF-7 cells, whereas no significant change was noted in MDA-MB-231 cells. Pyrosequencing analysis of breast cancer specimens demonstrated lower levels of average methylation in neighboring CpG sites compared to normal controls, confirming the association between hypomethylation and increased CEMIP expression in breast cancer (*Kuscu et al., 2012*). Additionally, studies suggest that TP53 mutation might lead to hypomethylation of the CEMIP promoter. TP53, commonly known as a tumor suppressor gene, when mutated, can impair DNMT1 function, a DNA methyltransferase that maintains DNA methylation patterns. Consequently, the CEMIP promoter might lose its methyl groups and become more active (*Hsieh et al., 2020*).

Beyond DNA methylation, histone modifications, mediated by hypoxia, also exert influence over transcription (*Watson et al., 2010*). Core histones stabilize DNA and regulate gene expression through various modifications, including methylation of lysine/arginine

residues (*Greer & Shi, 2012*; *Murray, 1964*). Aberrant histone methylation, particularly the loss of H3/H4 methylation and acetylation, characterizes tumor cells and profoundly impacts the expression of critical genes (*Albert & Helin, 2010*; *Chi, Allis & Wang, 2010*; *Greer & Shi, 2012*; *Yang & Bedford, 2013*). Histone H3 lysine 4 trimethylation (H3K4me3) at gene promoters serves to activate transcription. Invasive breast cancer cells displaying high CEMIP expression under hypoxia demonstrate enriched H3K4me3 at the CEMIP promoter, while less invasive cells exhibit lower H3K4me3 levels (*Evensen et al., 2015*). The demethylase JARID1A regulates H3K4me3 levels in hypoxic cells, with its overexpression reducing H3K4me3 at the CEMIP promoter (*Evensen et al., 2015*; *Zhou et al., 2010*).

Moreover, H3K27me3 modification deactivates gene transcription by condensing chromatin. The loss of H3K27me3 specifically characterizes aggressive subtypes of breast cancer and can serve as a valuable diagnostic marker. Epigenetic chemical screening has identified inhibition of histone H3K27me3 demethylation as a therapeutic strategy for triple-negative breast cancer (TNBC). Studies using functional analyses and RNA-seq/ChIP-seq data revealed that inactivating the protein CEMIP by increasing H3K27me3 resulted in reduced tumor cell growth and migration (*Hsieh et al., 2020*). Hence, methylation presents a multifaceted regulatory role in CEMIP transcription.

## POST-TRANSCRIPTIONAL REGULATION

Gene expression undergoes multiple levels of control, including post-transcriptional regulation following the initial transcription phase. Several mechanisms operate at this stage to govern gene expression. One significant process involves alternative splicing of pre-mRNA transcripts, enabling the generation of diverse protein isoforms from a single gene. RNA stability serves as another critical checkpoint—unstable mRNAs degrade swiftly, while stable ones persist longer for translation. RNA stability is modulated by RNA binding proteins and microRNAs, which identify specific sequences or structures within transcripts. MicroRNAs, in particular, bind to complementary regions in target transcripts, silencing gene expression by impeding translation or promoting mRNA decay. Additionally, RNA editing can modify the nucleotide sequence of RNA transcripts, often by converting adenosine to inosine. This process can alter protein-coding sequences or influence mRNA stability.

Regarding post-transcriptional regulation specific to CEMIP, current research predominantly centers on its regulation by microRNAs. The advancements in this field will be summarized to provide an overview of the progress in understanding how microRNAs regulate CEMIP.

### miR-29c-3p

The miR-29c-3p belongs to the miR-29 family, encompassing seven mature miRNAs, among them miR-29a-3p, miR-29b-3p, and miR-29c-3p. Prior investigations have indicated a frequent downregulation of miR-29 expression in cancer, with its diminished presence correlating with poorer overall survival (*Kwon et al., 2019*; *Qi et al., 2017*). One identified mode of action for miR-29c-3p as a tumor suppressor involves its indirect elevation of p53 signaling. This is achieved by miR-29c-3p targeting and suppressing the

PI3K subunits p85 α and CDC42, known as negative regulators of p53 (*Agarwal et al., 2015*; *Park et al., 2009*)

Further research has revealed the inhibitory effect of miR-29c-3p on cell migration and invasion by targeting KIF4A in ovarian cancer (*Feng et al., 2020*) and CEMIP in gastric cancer (*Wang et al., 2019a*). CEMIP, housing miR-29c-3p binding sites in its 3′UTR, experiences reduced expression upon miR-29c-3p overexpression, resulting in the suppression of gastric cancer cell migration (*Wang et al., 2019a*).

## miR-140-5p and 3p

The miR-140 gene encodes pre-miR-140, giving rise to two mature miRNAs, miR-140-5p and miR-140-3p. Located on chromosome 16q22.1, this gene resides within the 15th intron of the WW domain containing E3 ubiquitin protein ligase 2 (WWP2) human gene (*Toury et al., 2022*). Notably, miR-140-5p exhibits specific expression in cartilage (*Tuddenham et al., 2006*). Its involvement has been implicated in various disease contexts.

In retinoblastoma, miR-140-5p displays significant down-regulation in RB tissue, coinciding with an up-regulation of CEMIP expression (*Miao et al., 2018*). Elevated miR-140-5p levels reduce CEMIP expression, consequently weakening the proliferation, invasion, and migration abilities of RB cells (*Miao et al., 2018*). Functioning as a tumor suppressor in RB cells, miR-140-5p exerts its action by targeting CEMIP.

Meanwhile, miR-140-3p emerges as a potential biomarker for effective anti-tumor therapy. In cutaneous melanoma (CM), diminished miR-140-3p levels correlate with poorer CM survival rates, whereas overexpression of miR-140-3p impedes cell movement (*He et al., 2020*). Similarly, in gastric cancer, it suppresses the migration and invasion of gastric cancer cells by binding to SNHG12 and suppressing its expression (*Wang et al., 2021*).

In colorectal cancer (CRC), a negative correlation exists between miR-140-3p and CEMIP expression. Elevated CEMIP expression fosters the growth, colony formation, and invasion of CRC cells (*Yang et al., 2020*). Co-transfection experiments involving CEMIP and miR-140-3p in cancer cells demonstrated that miR-140-3p expression inhibits cancer cell growth, and the introduction of CEMIP reverses this inhibition (*Yang et al., 2020*). Studies indicate that miR-140-3p participates in CRC processes by binding to the CEMIP 3′UTR region and curbing its expression (*Yang et al., 2020*).

## miR-148-3p

miR-148a-3p is a part of the miR-148a family, plays crucial roles in regulating inflammation, immunity, and cancer progression. In laryngeal squamous cell carcinoma (LSCC), it targets DNA methyltransferase 1 (DNMT1), thereby influencing RUNX3 expression through DNMT1-mediated DNA methylation, effectively suppressing cell migration (*Jili et al., 2016*). Moreover, in gastric cancer, the expression of CEMIP shows an inverse relationship with miR-148a-3p levels. Reduced miR-148a-3p results in heightened CEMIP expression, promoting cell viability while impeding apoptosis. Investigations have confirmed CEMIP as a direct target of miR-148a-3p. The action of miR-148a-3p in regulating CEMIP portrays its role as a tumor suppressor in gastric cancer (*Song et al., 2021*). The downregulation of

miR-148a-3p consequently leads to the upregulation of CEMIP, potentially advancing gastric cancer. Thus, the miR-148a-3p/CEMIP axis represents a promising therapeutic target for intervention in gastric cancer progression.

### miR-188-5p

It is derived from non-coding RNA transcribed by the B cell integration cluster gene situated on chromosome 21, closely linked to various human tumors (*Wang et al., 2019b*). The miR-188 family includes two members: miR-188-3p and miR-188-5p. Studies have revealed abnormal expression of miR-188-5p in various pathological processes *in vivo*. Notably, miR-188-5p exhibits high expression levels in gastric cancer and liver cancer (*Tian et al., 2019*). In the context of multiple myeloma, miR-188-5p acts as a tumor suppressor by significantly curbing tumor cell proliferation and fostering apoptosis (*Liu et al., 2021a*).

In synovial fibroblasts' gene expression profile analysis, miR-188-5p exhibited significant regulatory influence on the hyaluronic acid binding protein CEMIP. Further exploration indicated the presence of a binding site for miR-188-5p in the CEMIP UTR sequence, implying direct regulation of this gene by miR-188-5p (*Ruedel et al., 2015*). This interaction slows down disease progression in which CEMIP plays a role to a certain extent.

### miR-216a

MicroRNA-216a (miR-216a) is closely associated with cancer and has been consistently found to be down-regulated in various cancers including breast cancer (*Cui et al., 2019*), renal cell cancer (*Wang et al., 2018*), colorectal cancer (*Zhang et al., 2017a*), lung cancer (*Zhen et al., 2018*), among others. Its tumor-suppressive role primarily involves the regulation of multiple signal pathways and transcription factors. For instance, in pancreatic tumors, miR-216a significantly curbs cell proliferation by targeting JAK2 and triggers programmed cell death in pancreatic tumor cells (*Hou et al., 2015*).

In the context of colorectal cancer, miR-216a has been implicated in the post-transcriptional regulation of CEMIP, influencing cancer development. The upregulation of CEMIP in colorectal cancer tissues promotes tumor invasion and correlates with a shorter survival period (*Zhang et al., 2017a*). Decreasing CEMIP expression in tumor cells leads to increased expression of epithelial markers (E-cadherin and cytokeratin 8, 18, 19) and decreased expression of mesenchymal markers (vimentin and Twist-1) (*Zhang et al., 2017a*). Conversely, CEMIP overexpression decreases cytokeratin expression and increases Twist-1 expression. These findings highlight CEMIP's role in promoting the epithelial-mesenchymal transition of colorectal cancer cells (*Zhang et al., 2017a*). As one of the direct functional targets of miR-216a, down-regulating CEMIP expression inhibits the migration and invasion of colorectal cancer cells *in vitro* and reduces the risk of metastasis *in vivo*.

### miR-296-3p

MicroRNA-296-3p (miR-296-3p) exhibits varying roles in different cancer developments. It is down-regulated in glioblastoma (*Bai et al., 2013*) and colorectal cancer (*Xiao & Li, 2023*),

while being up-regulated in squamous cell carcinoma (*Kakizaki et al., 2017*). MiR-296-3p contributes to the post-transcriptional regulation of CEMIP.

In the context of preeclampsia (PE), there's an abnormal increase in the mRNA level of miR-296-3p. Studies have identified a binding site for miR-296-3p on the CEMIP UTR. Interestingly, in cancer cells, CEMIP can counteract the down-regulation of $\beta$-catenin induced by miR-296-3p overexpression, shedding light on a potential mechanism whereby miR-296-3p might promote the PE process through its targeting of CEMIP (*Li et al., 2021b*).

## miR-486-3p and 5p

MicroRNA-486 (miR-486) originates from an intron within the ANK1 locus situated on chromosome 8p11.21. In mice, this locus harbors both miR-486a-5p and miR-486b-5p, transcribed in opposing directions but sharing the same mature sequence (*Douvris, Viñas & Burns, 2022*). MiR-486-3p displays tumor-suppressive characteristics, being downregulated in hepatocellular carcinoma (HCC) (*Ji et al., 2020*), retinoblastoma (*Yang et al., 2020*), osteosarcoma (*Zhang et al., 2021*), cervical cancer (*Ye et al., 2016*), adrenocortical carcinoma (*Li et al., 2021a*) ,and oral squamous cell carcinoma (*Garajei et al., 2022*). However, in glioblastoma (*Simionescu et al., 2022*), non-small cell lung cancer (*Jin et al., 2022*), and β-thalassemia (*Lulli et al., 2013*), it demonstrates an upregulation, thereby fostering disease progression.

In intervertebral disc degeneration (IDD) involving nucleus pulposus cells, miR-486-3p's expression negatively correlates with CEMIP. Experimental data affirms that miR-486-3p directly targets and downregulates CEMIP, influencing proliferation, apoptosis, and extracellular matrix synthesis in these cells. This sheds light on the clinical significance of the miR-486-3p/CEMIP axis for potential IDD intervention and therapy (*Cui & Zhang, 2020*).

On the other hand, miR-486-5p, a muscle-enriched miRNA found abundantly in plasma (*Bayés-Genis et al., 2018*; *Small et al., 2010*) , demonstrates differential expression patterns across various cancers. It is downregulated in lung cancer (*Moro et al., 2022*), steroid-induced osteonecrosis of the femoral head (SOFNH) (*Chen et al., 2023*), papillary thyroid carcinoma (*Ma et al., 2016*), breast cancer (*Li et al., 2019*) and non-small cell lung cancer serum (*Xing et al., 2020*), while being upregulated in cervical cancer (*Li et al., 2018*) and sepsis patient serum (*Sun & Guo, 2021*).

In non-small cell lung cancer (NSCLC), miR-486-5p interacts with CEMIP, impacting EGFR signaling and hampering NSCLC growth and metastasis upon overexpression. MiR-486-5p downregulates CEMIP, thus contributing to the promotion of NSCLC progression (*Wang et al., 2020*). The miR-486-5p/CEMIP axis provides crucial insights into understanding NSCLC's underlying mechanisms.

## miR-600

The expression of miR-600 undergoes dysregulation across various cancers. For instance, it is notably upregulated in prostate (*Valera et al., 2020*), ovarian (*Shan et al., 2022*) and several other cancers. In ovarian cancer, miR-600 facilitates proliferation and

metastasis by downregulating KLF9 (*Shan et al., 2022*). Additionally, in pancreatic ductal adenocarcinoma cells, miR-600 induces autophagy by targeting NACC1, thereby supporting proliferation (*Yang et al., 2021*). However, in breast tumors, low miR-600 expression correlates with active Wnt signaling and an unfavorable prognosis (*El Helou et al., 2017*). Remarkably, miR-600 can also exhibit tumor-suppressive behavior by inhibiting lung cancer through the downregulation of METTL3 (*Wei, Huo & Shi, 2019*). Furthermore, in colorectal cancer, miR-600 reduces matrix metalloproteinase-9 levels while increasing E-cadherin and β-catenin by negatively regulating p53 (*Zhang et al., 2017b*).

In colorectal cancer, a significant association exists between miR-600 and KIAA1199 (CEMIP). Notably, miR-600 displays downregulation in colorectal cancer tissues compared to normal tissues, while KIAA1199 mRNA exhibits significant upregulation. Lucidly demonstrated by luciferase reporter assays, miR-600 directly binds to the KIAA1199 3′UTR, solidifying KIAA1199 as a direct target of miR-600 (*Sun et al., 2018*).

### miR-4656

The expression of miR-4656 showcases dysregulation across various cancers (*Wang et al., 2019c*). In breast cancer, miR-4656 levels exhibit variability, being either upregulated or downregulated. Hepatocellular carcinoma cells demonstrate that ST8SIA6-AS1 regulates miR-4656, thereby upregulating HDAC11 to facilitate cell migration (*Fei et al., 2020*). Moreover, miR-4656 expression correlates with survival rates in glioma patients (*Shou, Gu & Gu, 2015*).

In prostate cancer, miR-4656 is typically downregulated while CEMIP is upregulated. Experimental overexpression of miR-4656 diminishes CEMIP mRNA levels and impedes cancer cell growth. Confirmatory studies established a binding site for miR-4656 within the CEMIP 3′UTR, affirming CEMIP as a direct target of miR-4656. Consequently, the downregulation of miR-4656 can result in upregulated CEMIP expression, fostering the progression of prostate cancer (*Hu & Lu, 2020*). In summary, miR-4656 demonstrates both tumor-suppressive and oncogenic roles across various cancer contexts, partly through its modulation of oncoproteins like CEMIP.

### miR-4677-3p

MiR-4677-3p, a recently identified microRNA, demonstrates varying expressions in different cancer types. In lung cancer, it's upregulated by circ_0001421, which acts as a tumor promoter. Interestingly, overexpressing miR-4677-3p inhibits proliferation and migration of lung cancer cells (*Zhang et al., 2020*). Conversely, in gastric cancer, miR-4677-3p is downregulated. Its restoration suppresses cell movement by directly binding to the UTR site of CEMIP, consequently inhibiting CEMIP expression. This inhibition leads to decreased proliferation, migration, and invasion of gastric cancer cells (*Mi et al., 2021*).

## CONCLUSIONS

CEMIP (cell migration inducing hyaluronan binding protein) emerges as a pivotal oncoprotein governing cancer advancement through diverse pathways. It orchestrates cell migration, invasion, and signaling cascades by influencing cytoskeletal dynamics, matrix

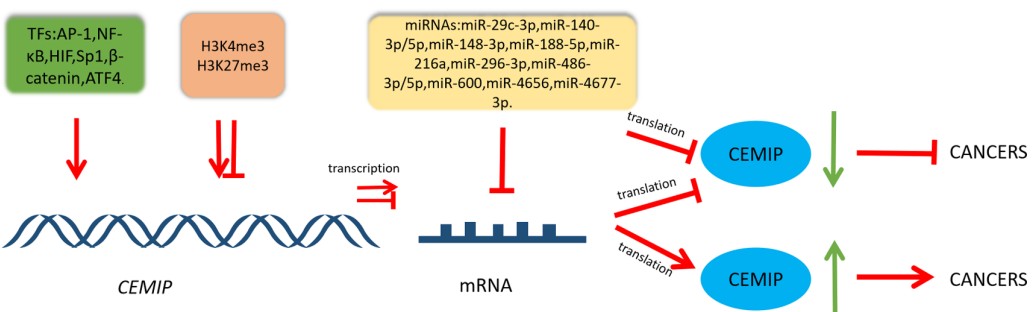

**Figure 2 Schematic diagram of CEMIP transcriptional and post-transcriptional regulation and its consequences for cancer development.** Transcription factors can enhance the transcription of CEMIP. This leads to promoting cancer development. CEMIP is also regulated by histone methylation. H3K4me3 in the CEMIP promoter region enhances gene transcription activity. On the other hand, H3K27me3 inhibits this process. miRNAs play an important role in CEMIP post-transcriptional regulation, which frequently leads to decreasing CEMIP and consequently inhibiting cancer development.

degradation, and crucial signaling pathways like ERK/MAPK. Moreover, its interaction with hyaluronan drives cell motility and adhesion while inducing epithelial-mesenchymal transition, a pivotal step in metastasis. Elevated CEMIP expression correlates with poor prognosis across various cancers.

While CEMIP holds promise as a potential target for early cancer diagnosis and therapy, it intersects with immune checkpoint blockade (ICB) therapy, mediating tumor immune escape. Targeting CEMIP may amplify the efficacy of ICB by boosting tumor immunogenicity and augmenting T cell activation and infiltration.

Studies employing CEMIP knockout mice elucidate its role in regulating inflammatory responses, wound healing, and infection. However, further optimization of these models is essential for more effective preclinical research. Efforts in drug development targeting CEMIP have shown promise, revealing its role in drug resistance and its potential as a biomarker for gastric cancer detection, prognostication, and treatment response.

CEMIP's expression is tightly controlled transcriptionally by a cadre of factors like AP-1, NF-κB, HIFs, and β-catenin, in concert with epigenetic modifications such as DNA methylation and histone marks. Furthermore, microRNAs wield post-transcriptional regulation over CEMIP expression, highlighting the intricate control mechanisms underpinning its oncogenic role across various cancers.

The intricate regulatory network that governs CEMIP highlights its pivotal role within diverse cellular pathways involved in cancer. This is visually represented in Fig. 2, while Table 1 provides a summary of its involvement in cancers, detailing both its transcriptional regulation and functional roles.

While recent studies have made headway in understanding CEMIP regulation, several critical questions beckon further exploration. Delving deeper into how signaling pathways like NF-κB, Wnt/β-catenin, and HIF-1 precisely modulate CEMIP expression, particularly in *in vivo* disease models, remains paramount. Expanding investigations across various cancer types might unveil distinct tissue-specific regulators governing CEMIP

**Table 1** CEMIP related to different cancers.

| Cancer | Expression | Targets | Mechanism | Biological function | Reference |
|---|---|---|---|---|---|
| Prostate cancer | High expression (mRNA/protein) | Sema3A | Activate the signal transduction of Sema3A | angiogenesis and cell metastasis, migration and invasion | *Luo et al. (2022)* |
| | | MMP2,VEGF, PDK4, LDHA | Activated by AMPK/GSK-3 β/ β-catenin cascade which participates in the process of anti-anoikis | | *Zhang et al. (2018)* |
| | | ITPR3 | Stablize ITPR3 expression to regulate $Ca^{2+}$ signal | cell detachment from matrix | *Liu et al. (2022)* |
| | | Bcl-2 | Induce Bcl-2-ser 70 phosphorylation | Autophagy, metastasis | *Yu et al. (2022)* |
| Colorectal cancer | High expression (mRNA/protein) | EphrinA2, ITPR3 | Stabilize the expression of EphrinA2 Stabilize the $Ca^{2}$ + signal involved by ITPR3 | cell proliferation and invasion | *Tiwari et al. (2013)* |
| | | EMT | Regulate Wnt/ β-catenin signal | cell metastasis | *Liang et al. (2018)* |
| | | β-catenin | Elevate O-GlcNAcylation of β-catenin and enhance β-catenin nuclear translocation from cytomembrane. | cell metastasis | *Hua et al. (2021)* |
| | | GRAF1 | Enhance the degradation of GRAF1 to activate CDC42/MAPK pathway-regulated EMT | cell metastasis | *Xu et al. (2023)* |
| Cholangiocarcinoma | High expression (mRNA/protein) | EMT | Mediate EMT through SMAD-independent pathway | metastasis, proliferation | *Zhai et al. (2020)* |
| Pancreatic ductal adenocarcinoma (PDAC) | High expression (mRNA) | HYAL1 | Not mentioned | cell growth ,migration | *Oba et al. (2021) Koga et al. (2017)* |
| | | HA | Promote the depolymerization of hyaluronic acid | migration | *Kohi et al. (2017)* |
| Small cell lung cancer | High expression (mRNA/protein) | HA | Depolymerization of high molecular weight hyaluronic acid | invasion, migration, metastasis | *Li et al. (2023)* |
| | | c-Myc | Inhibit the ubiquitination of c-Myc and increase the stability of c-Myc | cell multiplication | *Mo et al. (2023)* |
| Non-small cell lung cancer | High expression (mRNA/protein) | EMT | Mediate EMT through PI3K-AKT axis | Invasion, migration | *Tang et al. (2019)* |
| | | EGFR | Promote EGFR signaling and regulate EGFR-related molecules phosphorylation | Proliferation, migration | *Wang et al. (2020)* |

Guo et al. (2024), *PeerJ*, DOI 10.7717/peerj.16930

**Table 1** (*continued*)

| Cancer | Expression | Targets | Mechanism | Biological function | Reference |
|---|---|---|---|---|---|
| Hepatocellular carcinoma | High expression | EGFR,EMT | Promote EGFR phosphorylation and induce EMT process | metastasis, migration, invasion | *Xu et al. (2019)* |
| Gastric cancer | High expression (mRNA) | EMT,MMP | Enhance Wnt/ β-catenin signaling pathway | cell proliferation, migration, invasion | *Jia et al. (2017)* |
| Breast cancer | High expression (mRNA/protein) | Bip | Stabilize the expression of Bip | apoptosis, migration | *Banach et al. (2019)* |
| Ovarian cancer | High expression (mRNA/protein) | PI3K, AKT, AKT2 | Regulate the PI3K/AKT signal | cell proliferation, migration, invasion | *Shen et al. (2019)* |

**Notes.**

Sema3A, Semaphoring 3A; AMPK, AMP-activated protein kinase; GSK-3 β, Glycogen synthase kinase-3 β; MMP2, matrix metalloproteinase 2; VFGF, vascular endothelial-derived growth factor; PDK4, pyruvate dehydrogenase kinase isoform 4; LDHA, lactate dehydrogenase A; ITPR3, Inositol 1,4,5-trisphosphate receptor type 3; Bcl-2, B-cell lymphoma 2; EMT, Epithelial-mesenchymal transition; GRAF1, GTPase regulator associated with focal adhesion kinase-1; MAPK, Mitogen-activated protein kinases; HYAL1, hyaluronidase 1; HA, hyaluronidase; EGFR, Epidermal growth factor receptor; MMP, matrix metalloproteinase; PI3K, phosphoinositide 3-kinase.

transcriptional control. Extensive mapping of the CEMIP promoter and enhancers could unveil novel regulatory elements and binding sites for transcription factors, augmenting our understanding.

Epigenomic analyses represent an avenue to discover new methylation patterns or histone modifications influencing CEMIP expression, fostering a more comprehensive understanding of its regulation. Exploring the intricate interplay between transcription factors and epigenetic alterations is crucial. Further exploration is warranted to identify additional microRNAs or RNA binding proteins that target CEMIP, especially for their role in cancer pathogenesis and metastasis. Investigating mRNA stability, alternative splicing, or translational control mechanisms could shed light on post-transcriptional regulation. In essence, a nuanced comprehension of the multilayered transcriptional circuitry and post-transcriptional regulatory networks governing CEMIP expression holds promise in unveiling novel strategies to target this oncoprotein effectively in cancer therapeutics.

Presently, contradictory or inconclusive findings regarding CEMIP in cancers stem from its dual roles. Some studies suggest its association with tumor progression and aggressiveness, while others indicate its potential as a suppressor of cancer development. These discrepancies might stem from variations in experimental models, sample sizes, or contextual differences among different cancer types. Additionally, the diverse molecular functions of CEMIP across various cellular pathways could contribute to these conflicting observations, highlighting the need for further comprehensive investigations to elucidate its precise role in different cancer contexts.

## ACKNOWLEDGEMENTS

Thanks to all the members who participated in writing this reviews or provided help.

### Funding

This study was supported by the National Natural Science Foundation of China (81872005). The funders had no role in study design, data collection and analysis, decision to publish, or preparation of the manuscript.

### Grant Disclosures

The following grant information was disclosed by the authors:
National Natural Science Foundation of China: 81872005.

### Competing Interests

The authors declare there are no competing interests.

### Author Contributions

- Song Guo performed the experiments, analyzed the data, prepared figures and/or tables, authored or reviewed drafts of the article, and approved the final draft.
- Yunfei Guo analyzed the data, prepared figures and/or tables, and approved the final draft.

- Yuanyuan Chen analyzed the data, prepared figures and/or tables, and approved the final draft.
- Shuaishuai Cui analyzed the data, prepared figures and/or tables, and approved the final draft.
- Chunmei Zhang analyzed the data, prepared figures and/or tables, and approved the final draft.
- Dahu Chen conceived and designed the experiments, analyzed the data, prepared figures and/or tables, authored or reviewed drafts of the article, and approved the final draft.

## Data Availability

This is a literature review.

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
