# Peer review of "The role of CEMIP in cancers and its transcriptional and post-transcriptional regulation"

_PeerJ, doi:10.7717/peerj.16930_

## Round 0.1 · original submission · Minor Revisions

Please address the concerns of both reviewers and amend the manuscript accordingly.

**Language Note:** The review process has identified that the English language must be improved. PeerJ can provide language editing services - please contact us at copyediting@peerj.com for pricing (be sure to provide your manuscript number and title). Alternatively, you should make your own arrangements to improve the language quality and provide details in your response letter. – PeerJ Staff

Reviewer 1 ·

Basic reporting

This literature review by Song Guo et al. focuses on the role of Cell migration-inducing and hyaluronan-binding protein (CEMIP) in various cancers and the mechanisms controlling its expression. CEMIP, also known as KIAA1199 or HYBID, is a protein involved in cell migration and hyaluronic acid binding, playing a significant role in the breakdown of the extracellular matrix. The review highlights CEMIP's involvement in different cancers, linking it to diverse pathological states. It also delves into transcriptional and post-transcriptional mechanisms that modulate CEMIP levels, including the influence of transcription factors such as AP-1, HIF, and NF-κB, and the regulation by specific microRNAs. The review provides a comprehensive summary of current knowledge regarding CEMIP's role in cancer and its regulatory mechanisms.

Experimental design

1. While the review comprehensively summarizes the role of CEMIP in cancers, it might benefit from a deeper analysis of the molecular pathways involving CEMIP in specific cancer types. The connection between CEMIP expression and cancer progression, as well as potential therapeutic implications, could be explored in more detail.
2. Lack of Discussion on Contradictory Findings: The review does not sufficiently address contradictory or non-conclusive findings in the field. Acknowledging and discussing conflicting evidence would enhance the review's critical perspective.
3. The review could benefit from integrating data from different studies for a comparative analysis, which would provide a clearer understanding of CEMIP's role across various cancer types.
4. Some sections of the review, particularly those detailing the transcriptional and post-transcriptional regulation of CEMIP, would benefit from clearer explanations and delineation of the mechanistic pathways involved.

Validity of the findings

no comment

·

Basic reporting

The authors in this review offers an in-depth analysis of CEMIP's involvement in various cancer types, clarifying the ways in which transcriptional and post-transcriptional processes regulate its expression. Here are my comments:

In general, this review paper might use some more detail when addressing CEMIP and its relationship to various malignancies. Consider discussing the disease's downstream consequences and potential targets. When discussing the Wnt pathway (colon cancer), for example, elaborate on the downstream target genes that affect cell proliferation and migration. Such an illustration would improve readers' mechanistic knowledge of these molecular interactions.

Line 101, mention what are the implications of CEMIP related immune surveillance escape
Elaborate on the therapeutic potential of CEMIP knockout mice.

Provide further information about how increased CEMIP expression affects the malignant behavior of PDAC cells.

Explain how combining CA19-9 and CEMIP data enhances pancreatic cancer detection accuracy considerably. What is the rationale for this advancement, and how does it affect early detection or treatment planning?

Describe how CEMIP controls lymphocytes, immunosuppressants, immunostimulants, and MHC molecules in detail.

Discuss the connection between CEMIP expression and promoter hypomethylation in breast cancer cells. Give further insight on how this epigenetic regulation leads to CEMIP overexpression, as this is an important component of understanding the molecular processes at work.

Before getting into the specific role of CEMIP, briefly describe the general background of gastric cancer research and the existing level of knowledge.

Explain how CEMIP affects Wnt/-catenin signaling and MMP-mediated EMT specifically. This can help researchers better understand the molecular pathways that drive stomach cancer growth.

Discuss the therapeutic implications of targeting CEMIP in the development of novel gastric cancer medicines.

Line 175. Please provide a table showing the transcription factors regulated by CEMIP and also mention their downstream target genes and cancers associated. The same can be done for post transcriptional regulation.

Consider dividing the paragraph into subsections in the conclusion section for better organization, such as separating information on transcriptional and post-transcriptional regulation.

Experimental design

No comment

Validity of the findings

No comment

Additional comments

No comment

---

## Round 0.2 · accepted · Accept

Concerns of the reviewers were adequately addressed, and the revised manuscript is acceptable now.

Reviewer 1 ·

Basic reporting

The authors have addressed all of my concerns, making the paper suitable for publication in this journal.

Experimental design

The authors have addressed all of my concerns, making the paper suitable for publication in this journal.

Validity of the findings

The authors have addressed all of my concerns, making the paper suitable for publication in this journal.

Additional comments

no comments.